# Assessment of Diaphragm in Hemiplegic Patients after Stroke with Ultrasound and Its Correlation of Extremity Motor and Balance Function

**DOI:** 10.3390/brainsci12070882

**Published:** 2022-07-04

**Authors:** Xiaoman Liu, Qingming Qu, Panmo Deng, Yuehua Zhao, Chenghong Liu, Conghui Fu, Jie Jia

**Affiliations:** 1Department of Rehabilitation Medicine, Fudan University Huashan Hospital, Shanghai 200031, China; daqijingshenlai@126.com (X.L.); quqingming123@163.com (Q.Q.); 2National Clinical Research Center for Aging and Medicine, Fudan University Huashan Hospital, Shanghai 200031, China; 3Department of Rehabilitation Medicine, Jing’an District Central Hospital of Shanghai, Shanghai 200040, China; xyz3325127@163.com (P.D.); zyhsh0801@163.com (Y.Z.); 13817910015@163.com (C.L.); 4Shanghai Jinshan Zhongren Aged Care Hospital, Shanghai 201502, China; fuconghui1986@126.com; 5National Center for Neurological Disorders, Shanghai 200031, China

**Keywords:** diaphragm, stroke, extremity motor function, balance function

## Abstract

Background: A variety of functional disorders can be caused after stroke, among which impairment of respiratory function is a frequent and serious complication of stroke patients. The aim of this study was to examine diaphragmatic function after stroke by diaphragm ultrasonography and then to apply to explore its correlation with extremity motor function and balance function of the hemiplegia patients. Methods: This cross-sectional observational study recruited 48 hemiplegic patients after stroke and 20 matched healthy participants. The data of demographic and ultrasonographic assessment of all healthy subjects were recorded, and 45 patients successfully underwent baseline data assessment in the first 48 h following admission, including post-stroke duration, stroke type, hemiplegia side, pipeline feeding, pulmonary infection, ultrasonographic assessment for diaphragm, Fugl–Meyer Motor Function Assessment Scale (FMA Scale), and Berg Balance Scale assessment. Ultrasonographic assessment parameters included diaphragm mobility under quiet and deep breathing, diaphragm thickness at end-inspiratory and end-expiratory, and calculated thickening fraction of the diaphragm. The aim was to analyze the diaphragm function of hemiplegic patients after stroke and to explore its correlation with extremity motor function and balance function. Results: The incidence of diaphragmatic dysfunction under deep breath was 46.67% in 45 hemiplegia patients after stroke at the convalescent phase. The paralyzed hemidiaphragm had major impairments, and the mobility of the hemiplegic diaphragm was significantly reduced during deep breathing (*p* < 0.05). Moreover, the thickness fraction of hemiplegic side was extremely diminished when contrasted with the healthy control and non-hemiplegic side (*p* < 0.05). We respectively compared the diaphragm mobility under deep breath on the hemiplegic and non-hemiplegic side of patients with left and right hemiplegia and found there was no significant difference between the hemiplegic side of right and left hemiplegia (*p* > 0.05), but the non-hemiplegic side of right hemiplegia was significantly weaker than that of left hemiplegia patients (*p* < 0.05). The diaphragm mobility of stroke patients under quiet breath was positively correlated with age and FMA Scale score (*R*^2^ = 0.296, *p* < 0.05), and significant positive correlations were found between the diaphragm mobility under deep breath and Berg Balance Scale score (*R*^2^ = 0.11, *p* < 0.05), diaphragm thickness at end-inspiratory and FMA Scale score (*R*^2^ = 0.152, *p* < 0.05), and end-expiratory thickness and FMA Scale score (*R*^2^ = 0.204, *p* < 0.05). Conclusions: The mobility and thickness fraction of the hemiplegic diaphragm after stroke by diaphragm ultrasonography were significantly reduced during deep breathing. Diaphragm mobility on bilateral sides of the right hemiplegia patients were reduced during deep breathing. Moreover, the hemiplegic diaphragmatic function was positively correlated with extremity motor and balance function of the hemiplegia patients.

## 1. Introduction

Stroke is one of the leading causes of mortality and disability in China and worldwide [1]. A variety of functional disorders can be caused after stroke, among which impairment of respiratory function is a frequent and serious complication for stroke patients [2]. It is known that human respiratory function is regulated by both autonomic and volitional neural mechanisms. Automatic breathing is controlled by centers in the lower brain stem, whereas volitional breathing is controlled by the cerebral cortical centers [3]. Breathing can be activated volitionally or automatically via corticospinal and bulbospinal pathways, respectively [4]. Stroke disrupts multiple respiratory functions, and cortical damage results in decreased movement of the contralateral chest wall and diaphragm, while damage to the corticospinal pathway may lead to problems with ventilator drive, such as locked-in syndrome [5,6], and damage to the spontaneous breathing center in the brainstem leads to central sleep apnea [7]. The respiratory function of patients after stroke is significantly decreased, and the respiratory intensity is only about 50% of the normal population. The respiratory dysfunction can be attributed to the affected respiratory central nervous system [8] and respiratory muscles [9].

The main muscle of inspiration is the diaphragm, a thin, dome-shaped muscle positioned between the chest and abdomen. Rapidly conducting oligosynaptic pathways from motor cortex to the diaphragm were first demonstrated in person by Gandevia and Rothwell [10]. Maskill et al. used transcranial magnetic stimulation to confirm that the motor cortex of the diaphragm existed at a position approximately 3 cm lateral and 2 cm anterior to Cz, which particularly activated the contralateral inspiratory muscles [11]. The diaphragm is predisposed to atrophy because of central nervous system disorders [9]. Usually, the unilateral involvement of the diaphragm is paucisymptomatic in hemiplegic patients [12]. Therefore, diaphragm paralysis is under-diagnosed because of its varied and often non-specific presentation [13]. Early detection of diaphragm dysfunction is important for protecting patients from comorbid pulmonary problems [14].

Diaphragmatic dysfunction can be confirmed by a number of tests that mainly include chest radiograph [15], multi-slice spiral CT [16], magnetic resonance imaging (MRI) [17], sniff test [18], pulmonary-function test [14], respiratory muscle strength [19], transdiaphragmatic pressure (Pdi) [20], ultrasonography, electromyography (EMG) [21], transcranial magnetic stimulation (TMS) [11], etc. [22]. Ultrasonography of the diaphragm at its zone of apposition with the rib cage is a noninvasive technique. Ultrasonography can distinguish a functioning from a nonfunctioning diaphragm, so it can be used to diagnose both unilateral and bilateral diaphragmatic paralysis and to monitor recovery of the paralyzed diaphragm [22]. Ultrasound criteria for evaluation of normal and malfunctioning/paralyzed diaphragm have been published [23,24,25]. A study [26] reviewed the normal and pathologic values for diaphragm ultrasound; the normal values of diaphragmatic tidal excursion, which were 16 ± 3 mm in women and 18 ± 3 mm in men; the normal values of diaphragm mobility (deep breathing), which were 57 ± 10 mm in women and 70 ± 11 mm in men, while normal value of diaphragm thickness was 2.7 ± 0.5 mm, and thickening fraction was 37 ± 9% [24], and the criteria for diaphragm thickening fraction ≤ 20% or diaphragm thickness ≤ 2 mm with inspiration can evaluate the dysfunction of the diaphragm [22,23]. Ultrasonography is portable, ubiquitous in medical facilities, has no risk of ionizing radiation, and it carries the advantage of assessing the diaphragm at the bedside. Diaphragmatic ultrasound allows both morphologic assessment and functional evaluation of the muscle that can be used to measure changes in the thickness and motion of the diaphragm during inspiration. Furthermore, it allows repeated measurements over time [5].

Balance is essential for performing everyday activities [27]. The diaphragm can stabilize the trunk and spine during activities [28]. One study investigated the effects of diaphragm training on balance ability in subjects with hemiplegia due to stroke [29] and found that diaphragm training could lead to improved static stability and dynamic balance [30]. However, balance impairment was associated with reduced strength [31]. Although certain studies have demonstrated that there is a relationship between diaphragm and pulmonary function [32], swallowing function [33,34], trunk control [9,35], and respiratory muscular strength, no study has examined its correlation with extremity motor function, balance function of hemiplegic side. In this context, the present study has two objectives. The first is to examine diaphragmatic function after stroke by diaphragm ultrasonography. The second objective is to apply to explore its correlation with extremity motor function and balance function.

## 2. Materials and Methods

### 2.1. Participant Selection

This cross-sectional observational study (Appendix A) enrolled 48 patients with stroke from the Department Rehabilitation Medicine of Fudan University Huashan Hospital and of Jing’an Branch of Fudan University Huashan Hospital from October 2021 to February 2022. This study also included 20 age-matched healthy subjects. The study was approved by the ethics committee of the institution (No. 2021/1004) and was prospectively registered at the Chinese Clinical Trial Registry (Chi-CTR-2200056309). All patients provided written informed consent prior to enrollment.

Inclusion criteria in the study were as follows: inclusion criteria: (1) a radiological and clinically diagnosis of ischemic or hemorrhagic stroke within 1–6 months from the onset, (2) between 30 and 80 years old, (3) unilateral hemiplegia, and (4) good listening comprehension and can follow instructions. The exclusion criteria were as follows: (1) cognitive dysfunction (MoCA Scale [36] score < 26 points); (2) unable to complete the assessment due to deficits in hearing, vision, or understanding; (3) history of a known restrictive and/or obstructive pulmonary disease or cardiac disease; (4) history of previous thoracic and/or abdominal surgery; and (5) history of excessive alcohol consumption. Healthy subjects were screened and recruited as follows: We selected the age-matched healthy participants with normal lung function (FEV1 > 80% pred and FVC > 80% pred) according to the age of stroke participants at the ratio of 2:1, and the exclusion criteria of the healthy subjects was same to the hemiplegic patients. The ratio between hemiplegic patients and healthy subjects was calculated based on the previously reported data, and the final target sample size was with a ratio of 1:1~3:1 [37,38].

### 2.2. Study Protocol

The participants were evaluated with the diaphragm mobility and thickness with ultrasound under quiet and deep breathing, the general characteristics, and the assessments of extremity motor and balance function.

### 2.3. Assessments

#### Ultrasonographic Assessment for Diaphragm

In general, there were two major forms of ultrasonographic assessment of the diaphragm: diaphragm excursion and thickness [39]. The feasibility and reliability of the measurements have been established in healthy subjects and patients [24,40,41]. Ultrasonographic technique was previously reported to be reliable, with high intra-class correlation coefficient for intra-rater and inter-rater reliability [23]. In this study, all sonographic examinations were performed by the same experienced radiologist using a SONIMAGE HS1 ultrasound machine (KONICA MINOLTA, Tokyo, Japan). The radiologist was blinded to the presence or side of hemiplegia. Ultrasonography examinations were carried out 2–3 h after a meal. After the patients were allowed to rest for 5–10 min [42].

Diaphragm mobility measurement: With all the participants in the supine position, B-mode ultrasonography (Figure 1a) with a low-frequency sector transducer (2–5 MHz) was used to measure the diaphragm mobility during respiratory. With the probe angled cranially, the liver was used as a window on the right side of the diaphragm, and the diaphragm was examined from the anterior subcostal approach by positioning the probe below the right costal margin between the midclavicular and anterior axillary lines (Figure 1d). While the spleen was used for the left side of the diaphragm [42], the diaphragm was examined from a subcostal or low intercostal approach by positioning the probe between the anterior axillary and midaxillary lines [24] (Figure 1d). To control for potential bias, for a given individual, the mobility of the bilateral diaphragm in the same position were recorded by M-mode ultrasonography (Figure 1b) during several respiratory cycles. Three or more respiratory cycles were recorded in each subject, respectively, during quiet breathing and deep breathing (Figure 1e). The distance between the echogenic lines was measured in frozen images, and measurements were averaged from at least three different cycles.

Diaphragm thickness measurement: With all the study participants in the supine position, a high-frequency linear array transducer (10–14 MHz) was used to measure the diaphragm thickness at the zone of apposition during inspiration or expiration. The probe was positioned on the chest wall at approximately the anterior axillary line at the eighth and ninth intercostal spaces [14] (Figure 1d). With the probe perpendicular to two ribs, the diaphragm can be visualized as a three-layered structure consisting of a relatively non-echogenic muscular layer bounded by echogenic membranes of the peritoneum and diaphragmatic pleura [39]. The thickness of the bilateral diaphragm in the same position were recorded by B-mode (Figure 1f,g) or M-mode (Figure 1c) ultrasonography during several respiratory cycles. Three images were recorded in each subject, respectively, at end-inspiration (Figure 1g) and end-expiration (Figure 1f) during deep breathing. The participants were asked to breathe in as deeply as they possibly could. Thickening fraction (TF) reflects on tractile activity that can be used to assess muscle function [40,43]. TF (%) = (end-inspiratory thickness − end-expiratory thickness) / end-expiratory thickness × 100%. Diaphragm thickness was analyzed using imaging software, Image-J 1.8.0 (National Institutes of Health, Bethesda, MD, USA). The measurements were performed three times and expressed as the mean.

### 2.4. Assessment of Other Parameters

The general characteristics of the subjects were recorded, including gender, age, smoking, post-stroke duration, stroke type, hemiplegia side, pipeline feeding, and pulmonary infection. The following assessments were performed, including Fugl–Meyer Motor Function Assessment Scale (FMA Scale) and Berg Balance Scale. All hemiplegic patients received assessments of extremity motor function and balance function through administration of the FMA Scale and Berg Balance Scale by an experienced physician from the rehabilitation medicine department to control potential bias. FMA Scale was one of the most used and recommended assessment scales of sensorimotor function in stroke; the item, subscale, and total score level reliabilities were high, and the scale could be recommended for use in general [44]. The Berg Balance Scale has been confirmed to be an effective way of balance measurement in patients with stroke; the inter-rater, intra-rater, and examined test–retest reliability were excellent [45].

### 2.5. Sample Size Calculation

A calculation of sample size was performed using PASS software. Sample size was calculated using data from our previous study, which showed the difference between the average mobility of the two groups was 0.5. We calculated that at least 34 patients were needed to provide the study with a sufficient statistical power of 0.9 and an alpha of 0.05.

### 2.6. Statistical Analysis

Statistical analyses were conducted using SPSS 23.0 (IBM Corporation, Armonk, NY, USA). Continuous data are presented as the mean ± SD (standard deviation) or median, and frequencies were calculated for categorical variables. The normality of data distribution and residuals from linear regressions were evaluated using the Shapiro–Wilk normality test. The differences between the groups were evaluated using the independent samples *t*-tests. Non-parametric tests were conducted when the values did not follow a normal distribution, differences between conditions were analyzed with a paired Wilcoxon rank test, and differences between groups were evaluated with non-paired tests. Categorical variables were analyzed with the chi-square test. One-way ANOVA or Kruskal–Wallis test analyzed multi-group comparison. Correlations were tested with the Pearson correlation coefficient. Multiple linear regression analysis to evaluate correlation. Significance was set at 0.05. The value of *p* < 0.05 was considered significant, with symbols presenting as * for *p* < 0.05, ** for *p* < 0.01, and *** for *p* < 0.001.

## 3. Results

### 3.1. Demographic Data

All the recruited healthy participants and 45 hemiplegic patients successfully underwent the whole procedure of interview, assessment, and measurement, while 3 of the 48 recruited patients withdrew from the study because of the failure of left diaphragmatic mobility measurement, and we excluded the 3 patients who had data missing. All patients’ data collection was completed in the first 48 h following admission. The details of demographic and indicators related are displayed in Table 1. We matched the age of the included participants. The average post-stroke duration was 3.33 ± 1.71 months, and 29 of them were ischemic, while 16 of them were hemorrhagic; the proportion was, respectively, 64.44% and 35.56%. The left hemiplegia was 27 with a proportion of 60%, while right hemiplegia was 18 with a proportion of 40%. The incidence of those requiring pipeline feeding and pulmonary infection were, respectively, 11.11% (*n* = 5) and 15.56% (*n* = 7). The overall incidence of diaphragmatic dysfunction under deep breathing was 46.67% (*n* = 21) in 45 patients with stroke, and diaphragmatic dysfunction was seen on the hemiplegic side 40% (*n* = 18) vs. 17.78% (*n* = 8) on non-hemiplegic side, with a statistically significant difference (*p* = 0.035 < 0.05).

### 3.2. Diaphragmic Data by Ultrasonography

We compared the differences of the mobility, thickness, and thickening fraction of the bilateral diaphragm by ultrasonography between 20 healthy subjects and 45 hemiplegic patients. In Table 2, all the diaphragms were divided into three groups: healthy control (*n* = 40), hemiplegic side (*n* = 45), and non-hemiplegic side (*n* = 45). There were no statistically significant differences between the three groups in mobility under quiet breathing and thickness at end-inspiration and end-expiration under deep breathing (*p* > 0.05). The results of the comparison between groups can be found in the Figure 2. The mobility of bilateral diaphragm in hemiplegic patients was weaker than that in healthy participants during deep breathing (*p* < 0.05). The mobility of the hemiplegic diaphragm was significantly reduced during deep breathing (*p* < 0.05). Moreover, the thickness fraction of hemiplegic side was extremely diminished when contrasted with the healthy control and non-hemiplegic side (*p* < 0.05). There was no statistically significant difference between bilateral diaphragm of the healthy control and non-paralyzed hemidiaphragm in thickness fraction (*p* > 0.05).

As shown in Table 3 and Figure 3, there were no statistically significant differences between the left and right diaphragm in mobility under quiet breathing, thickness at end-inspiration, and end-expiration under deep breathing of the three groups as well as the thickening fraction of the healthy control (*p* > 0.05). The right diaphragm mobility of healthy control during deep breath was significantly better than that of the left side (*p* < 0.05). Whether it was left hemiplegia or right hemiplegia, the thickening fraction of the diaphragm on the hemiplegic side was significantly lower than that on the non-hemiplegic side (*p* < 0.05), while the mobility of the diaphragm on the hemiplegic side in the left hemiplegia patients during deep breath were significantly weaker than that on the non-hemiplegic side (*p* < 0.05). There was no significant difference in the mobility of the bilateral diaphragm in the right hemiplegia patients during deep breath (*p* > 0.05) (Figure 3). We respectively compared the diaphragm mobility under deep breath on the hemiplegic and non-hemiplegic side of patients with left and right hemiplegia and found there was no significant difference between the hemiplegic side of right and left hemiplegia (*p* > 0.05), but the non-hemiplegic side of right hemiplegia was significantly weaker than that of left hemiplegia patients (*p* < 0.05) (Table 4).

### 3.3. The Correlation between Diaphragm Data and Other Parameters

The results of Pearson correlation coefficient analysis were shown in Figure 4. Significant positive correlations were found between diaphragm mobility under quiet breath and age (*r* = 0.40, *p* = 0.006) and FMA score (*r* = 0.46, *p* = 0.002); diaphragm mobility under deep breath and Berg Balance Scale score (*r* = 0.33, *p* = 0.032); diaphragm thickness in end-inspiratory and FMA score (*r* = 0.39, *p* = 0.012); and diaphragm thickness in end-expiratory and FMA score (*r* = 0.45, *p* = 0.003). In addition, significant positive correlations were found between diaphragm mobility under quiet breath and under deep breath (*r* = 0.39, *p* = 0.008); diaphragm thickness in end-inspiratory and in end-expiratory (*r* = 0.72, *p* = 0.000) and thickening fraction (*r* = 0.62, *p* = 0.000); and FMA score and Berg Balance Scale score (*r* = 0.63, *p* = 0.000).

The correlations between the diaphragm data measured by ultrasound and the demographic data, extremity motor function, and balance function variables are shown as a correlation matrix in Figure 5. Table 5 shows the results of multiple linear regression analysis coefficients. The diaphragm mobility of stroke patients under quiet breath was positively correlated with age and FMA score (*R*^2^ = 0.296, *p* < 0.05), and significant positive correlations were found between the diaphragm mobility under deep breath and Berg Balance Scale score (*R*^2^ = 0.11, *p* < 0.05), diaphragm thickness in end-inspiratory and FMA score (*R*^2^ = 0.152, *p* < 0.05), and end-expiratory thickness and FMA score (*R*^2^ = 0.204, *p* < 0.05).

## 4. Discussion

The criteria for diaphragm thickening fraction ≤20% can evaluate the dysfunction of the diaphragm. Catalá-Ripoll JV et al. used the criterion to observe the incidence of diaphragmatic dysfunction in acute ischemic stroke, and the incidence was 51.7% under normal breathing and 11.5% under forced breathing [46]. In this study, 21 of 45 stroke patients had diaphragmatic dysfunction, and the incidence was 46.67%. Considering the reasons, on the one hand, there were certain individual differences in the innervation of the phrenic nerve; on the other hand, the average duration of stroke in this study was 3 months, which was mainly in the convalescent period. However, the incidence of diaphragmatic dysfunction in different phases of stroke has not been reported. Eric et al. showed that diaphragmatic strength after unilateral diaphragmatic paralysis seems to improve with time [37], while on the contrary, McCool et al. [23] found that with the prolongation of the duration of the disease, the lower the diaphragm thickening fraction in chronic stroke patients, and the greater the possibility of atrophy of the diaphragm. Therefore, long-term inspiratory muscle training strategies may be more beneficial to the maintenance and retrieval of respiratory function in stroke patients. We also observed the incidence of diaphragmatic dysfunction on the hemiplegic and non-hemiplegic sides, respectively, at 40% and 17.78%, which confirmed that diaphragm was mainly innervated by the contralateral nerve, while certain proportions of diaphragmatic dysfunction were found in the non-hemiplegic side, which verified that the diaphragm was partially innervated by the ipsilateral nerve.

The results of intra-group comparison showed no differences between bilateral diaphragm of healthy subjects and of hemiplegia patients in diaphragm mobility, diaphragm thickness at end-inspiration, and end-expiration during quiet breathing. In the present study, we also found no reduction in diaphragmatic excursion of hemiplegic side during quiet respiration. Automatic breathing is controlled by centers in the lower brain stem, which might be explained by the fact that it is primarily an automatic respiration pattern under quiet breathing. Meanwhile, compared with healthy control and non-hemiplegic side, we found a significant reduction in diaphragm mobility and thickening fraction of hemiplegic side during deep respiration, which is consistent with the result of previous research [35]. Lesions in the hemisphere might lead to diaphragmatic dysfunction contralateral to the lesion. Therefore, the functional rehabilitation strategies after stroke should focus on the diaphragm function of the hemiplegic side. As described in the present paper, during deep inspiration, there was a significant bilateral reduction in hemispheric diaphragmatic excursion in patients with stroke for both the hemiplegic and non-hemiplegic side when compared with healthy subjects. However, the paralyzed hemidiaphragm had major impairments, and the mobility of the hemiplegic diaphragm was significantly reduced during deep breathing. Considered among the reasons is that, firstly, diaphragmatic paresis is mainly contralateral to the cerebral lesion [47]. Secondly, the diaphragm is both contralaterally innervated and ipsilaterally innervated [48], and innervation exhibits marked variations from person to person [2]. Thirdly, taking into account the fact that the paralyzed hemidiaphragm usually has a pendulum movement into the thorax (paradoxical movement) [22], it has the potential to impair the pressure generated during inspiration even in the non-paralyzed hemidiaphragm [38]. Some previous studies had different considerations. Verin et al. believed that the non-paralyzed hemidiaphragm increased in strength to compensate for the dysfunction of the paralyzed hemidiaphragm [25]. However, Clare et al. thought this was not true for all patients [49]. Similar to our findings, Cohen et al. [3] found that patients with hemiplegia had reduced diaphragm motion during voluntary inspiration on the same side of the body paralysis, and this finding was not seen during quiet respiration. Houston et al. [47] also found bilaterally decreased volitive diaphragmatic motion in acute cerebral infarction. Thus, for hemiplegic patients, more attention should be paid to the bilateral diaphragm function. In addition, the thickening fraction on the hemiplegic side was significantly lower than that in the healthy control and the non-hemiplegic side, while there was no significant difference between the non-hemiplegic side and the healthy subjects.

As for the comparisons of ultrasound parameters of left and right diaphragm in healthy subjects, we found a significant difference in left and right diaphragm mobility during deep breathing. However, there was no significant difference between left and right diaphragm in the mobility during quiet breathing as well as the thickness and thickening fraction. Contrary to what has been reported in the literature [50], we found healthy subjects had a larger excursion on the right side than on the left side. Comparison of diaphragm on different hemiplegic sides showed that the mobility of the diaphragm on the hemiplegic side in the left hemiplegia patients during deep breath were significantly reduced than that on the non-hemiplegic side, and we also found there was no significant difference between the bilateral diaphragm mobility in the right hemiplegia patients during deep breathing. Then, we respectively compared the diaphragm mobility on the hemiplegic and non-hemiplegic sides of patients with left and right hemiplegia and found that there was no significant difference between the hemiplegic side of right and left hemiplegia, but the non-hemiplegic side of right hemiplegia was significantly weaker than that of left hemiplegia patients. The study of Kang-Jae Jung et al. [32] found that the movement of the diaphragm on both sides of the patients with right hemiplegia was reduced, which was similar to the results of this study. Therefore, for patients with right hemiplegia, more attention should be paid to the early detection and intervention of diaphragmatic dysfunction and the rehabilitation of bilateral diaphragmatic function. There may exist an interhemispheric difference of the centrally innervated diaphragm. However, there is no published research on the dominant center of the brain innervated by the phrenic nerve, and it is expected to further explore its mechanism by using transcranial magnetic stimulation (TMS), functional magnetic resonance imaging (fMRI), etc.

Unilateral paralysis, which consequently could affect the diaphragm and possible postural deviations in the trunk, could trigger respiratory changes in these patients [51]. One study showed that chronic stroke survivors had decreased abdominal muscle thickness on the affected side, and respiratory muscle function had positive correlation with trunk function and balance [35]. In this study, we analyzed the correlation between diaphragm ultrasonographic parameters and extremity motor function and balance function. The results showed that the mobility of the diaphragm under quiet breathing on the hemiplegic side in stroke patients was positively correlated with age and positively correlated with FMA Scale score. The mobility of the diaphragm under deep breathing on the hemiplegic side was positively correlated with Berg Balance Scale score, and the thickness of the diaphragm at the end of inspiration and expiration was positively correlated with FMA Scale score. Multiple linear regression analysis showed that there was a positive correlation between the diaphragm function and extremity motor function and balance function of the hemiplegic side. However, the relationship did not imply causality. Further research is needed for the causation. In recent years, our research group has studied the use of occupational therapy combined with pulmonary function training in the treatment of stroke patients, which was more beneficial to the recovery of upper extremity function. Hence, it may provide more options for the rehabilitation of patients with extremity motor and balance dysfunction.

Several limitations existed in our study. Firstly, the sample size of the healthy control group was relatively small. Our study sample comprised 45 patients, and the sample may not be sufficient to obtain the incidence of diaphragmatic dysfunction, but as a preliminary study, it can reveal part of the incidence. Future studies with larger sample sizes are necessary to confirm the findings of this study. Secondly, it was a limitation that the average duration of stroke in this study was 3 months, which was mainly in the convalescent period within 1–6 months from the onset. There was a mismatch in control for the ratio of males to females at enrollment, and the inclusion criteria of the subjects was a wide age range. Thirdly, the measurement of diaphragm thickness and the calculation of diaphragm thickening fraction in this study are based on deep breathing, and the results may be different due to the breathing effort of each individual. This subsequent reproducibility is a limitation common in many of the studies. However, in this study, we used the same observer for three different cycles of measurement and used a unified standard for measurement software, thus eliminating within-observer variability.

## 5. Conclusions

The incidence of diaphragmatic dysfunction under deep breath was 46.67% in 45 hemiplegia patients after stroke at convalescent phase. The mobility and thickness fraction of the hemiplegic diaphragm after stroke by diaphragm ultrasonography were significantly reduced during deep breathing. Diaphragm mobility on bilateral sides of the right hemiplegia patients were reduced during deep breathing. Moreover, the hemiplegic diaphragm function was positively correlated with extremity motor and balance function of the hemiplegia patients.

## Figures and Tables

**Figure 1 brainsci-12-00882-f001:**
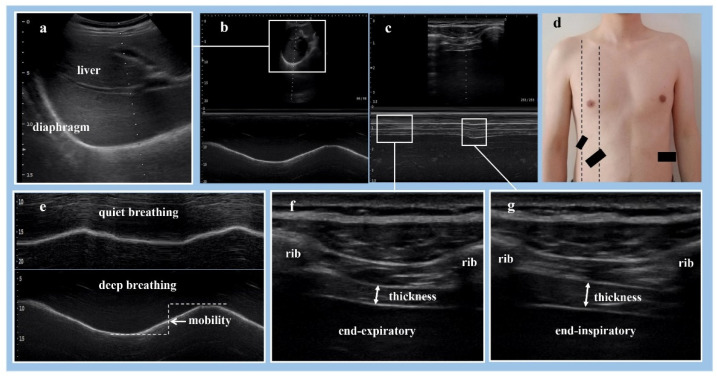
(**a**) B-mode ultrasonography, where the hyperechoic line is the diaphragm, and the dotted line has an angle of about 30°. (**b**) M-mode ultrasonography, where the movement amplitude of the hyperechoic line is the mobility of the diaphragm. (**c**) M-mode ultrasonography, where the dotted line is positioned on the diaphragm, and diaphragm thickness can be observed during breathing. (**d**) The probes position, where the left dotted line is the anterior axillary line, and the right dotted line is the midclavicular line, and the two larger black boxes are low frequency probes, while the smaller black box is the high-frequency probe. (**e**) M-mode ultrasonography showing the measurement of diaphragm mobility during quiet and deep breathing. (**f**,**g**) B-mode ultrasonography showing the measurement of diaphragm thickness at end-inspiration and end-expiration.

**Figure 2 brainsci-12-00882-f002:**
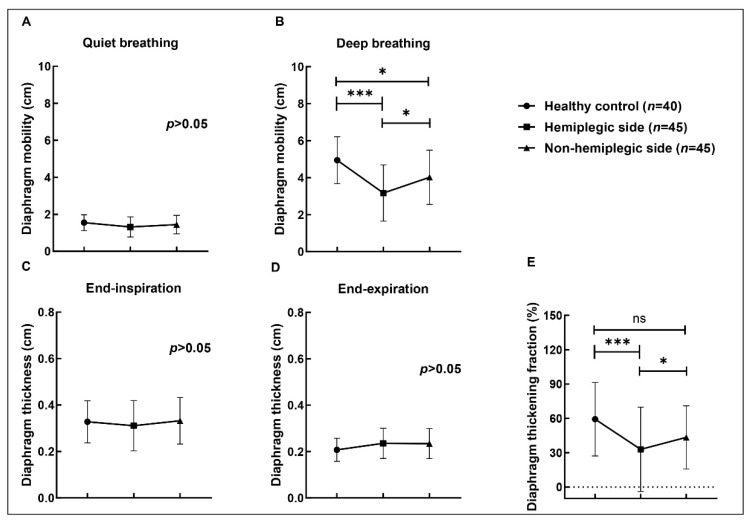
Diaphragm mobility, thickness, and thickening fraction data of intra-group comparison. (**A**,**B**) were expressed the mobility measured under quiet and deep breathing respectively. End-inspiratory thickness expressed in (**C**), end-expiratory thickness expressed in (**D**), and thickening fraction (%) expressed in (**E**) were measured under deep breathing. The value of *p* < 0.05 was considered significant, with symbols presenting as * for *p* < 0.05 and *** for *p* < 0.001.

**Figure 3 brainsci-12-00882-f003:**
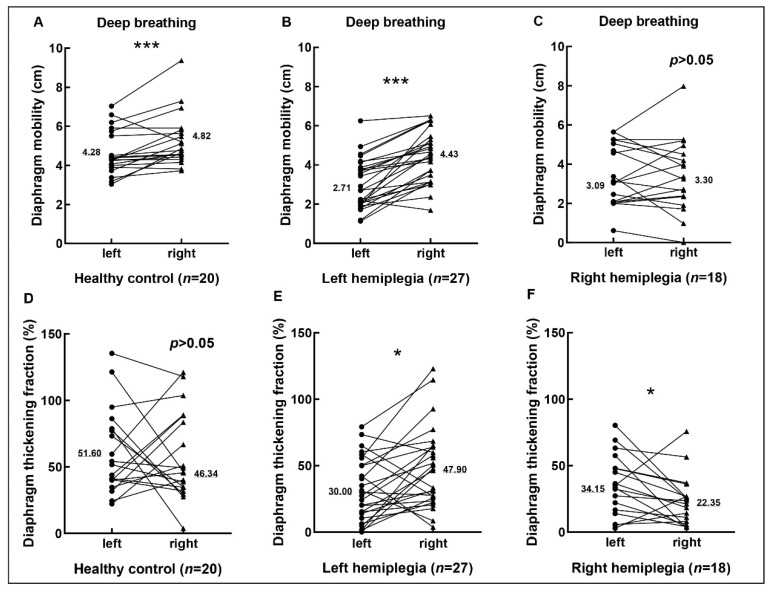
The comparison of left and right diaphragm mobility and thickening fraction of healthy control (**A**,**D**), left hemiplegic (**B**,**E**), and right hemiplegic (**C**,**F**) patients. Diaphragm mobility and thickening fraction expressed were measured under deep breathing. The values shown in each graph are the median. The value of *p* < 0.05 was considered significant, with symbols presenting as * for *p* < 0.05 and *** for *p* < 0.001.

**Figure 4 brainsci-12-00882-f004:**
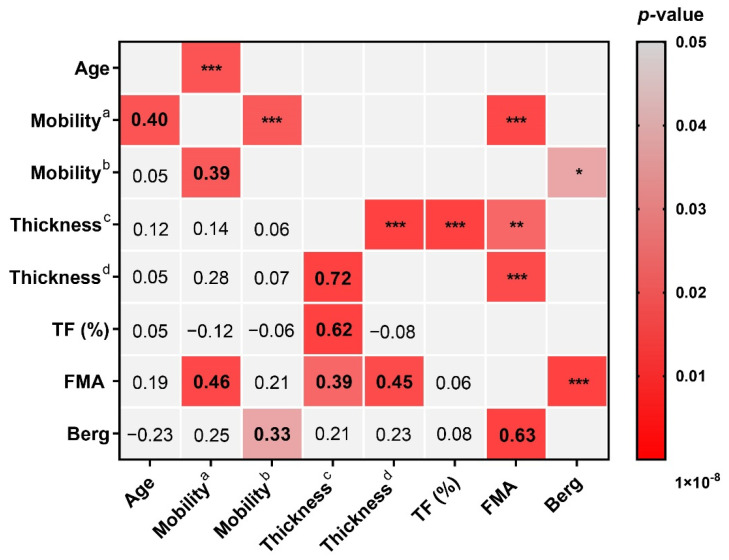
The correlation between diaphragm ultrasound parameters and the demographic data, extremity motor function, and balance function using Pearson correlation coefficient analysis. The red grids in the figure represent correlations, and the gray grids represent non-correlations. The values in the grids are r values, and positive values indicate positive correlations; conversely, negative values indicate negative correlations. The * in the grids represent *p*-value; the value of *p* presents as * for *p* < 0.05, ** for *p* < 0.01, and *** for *p* < 0.001. Mobility^a^, mobility (quiet breath); Mobility^b^, mobility (deep breath); Thickness^c^, thickness (end-inspiratory); Thickness^d^, thickness (end-expiratory); TF (%), thickening fraction. FMA, Fugl–Meyer Motor Function Assessment Scale score; Berg, Berg Balance Scale score.

**Figure 5 brainsci-12-00882-f005:**
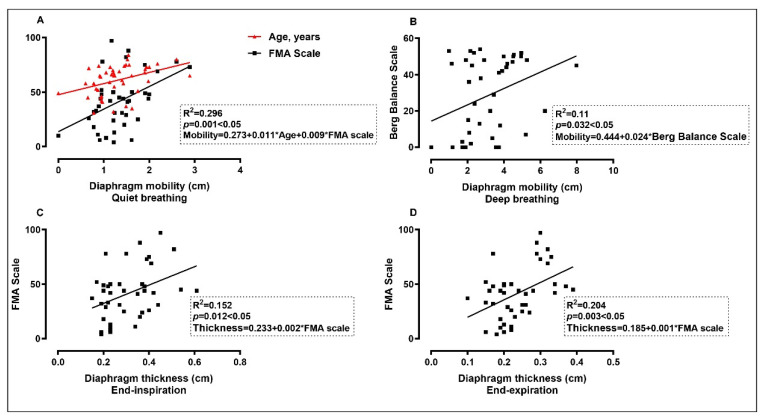
The correlation between diaphragm ultrasound parameters and the demographic data, extremity motor function, and balance function using multiple linear regression analysis. FMA Scale, Fugl–Meyer Motor Function Assessment scale. * *p* < 0.05.

**Table 1 brainsci-12-00882-t001:** General characteristics of two groups.

	Healthy Subjects (*n* = 20)	Hemiplegic Patients (*n* = 45)	*p*
Gender, *n* (%)			0.001 ***
Male	8 (40)	37 (82.22)	
Female	12 (60)	8 (17.78)	
Age, years (mean ± SD)	55.95 ± 11.75	61.02 ± 13.66	0.155
Smoking, *n* (%)			0.356
Yes	4 (20)	14 (31.11)	
No	16 (80)	31 (68.89)	
Post-stroke duration, months (mean ± SD)	-	3.33 ± 1.71	-
Stroke type, *n* (%)			-
Ischemic	-	29 (64.44)	
Hemorrhagic	-	16 (35.56)	
Hemiplegic side, *n* (%)			-
Left	-	27 (60)	
Right	-	18 (40)	
Pipeline feeding, *n* (%)			-
Yes	-	5 (11.11)	
No	-	40 (88.89)	
Pulmonary infection, *n* (%)			-
Yes	-	7 (15.56)	
No	-	38 (84.44)	
Diaphragmatic dysfunction, *n* (%)			0.000 ***
Yes	1 (5)	21 (46.67)	
No	19 (95)	24 (53.33)	

Data of age and post-stroke duration expressed as mean ± SD; *n* (%) for gender, smoking, stroke type, hemiplegic side, pipeline feeding, pulmonary infection, and diaphragmatic dysfunction. SD, standard deviation. The value of *p* < 0.05 was considered significant, with symbol presenting as *** for *p* < 0.001.

**Table 2 brainsci-12-00882-t002:** Diaphragm mobility, thickness, and thickening fraction data of healthy control, hemiplegic, and non-hemiplegic side.

	Healthy Control (*n* = 40)	Hemiplegic Side (*n* = 45)	Non-Hemiplegic Side (*n* = 45)	*p*
Mobility (quiet breath)	1.54 ± 0.44	1.31 ± 0.54	1.44 ± 0.50	0.105
Mobility (deep breath)	4.95 ± 1.27	3.17 ± 1.52	4.02 ± 1.47	0.000 ***
Thickness (end-inspiratory)	0.33 ± 0.09	0.31 ± 0.11	0.33 ± 0.10	0.378
Thickness (end-expiratory)	0.21 ± 0.05	0.24 ± 0.07	0.23 ± 0.06	0.072
TF (%)	59.29 ± 32.11	32.95 ± 36.85	43.42 ± 27.62	0.000 ***

Data expressed as mean ± SD. TF (%), thickening fraction (deep breath). *p*-values for differences in healthy control versus hemiplegic side versus non-hemiplegic side, analyzed using the one-way ANOVA or Kruskal–Wallis analysis. The value of *p* < 0.05 was considered significant, with symbol presenting as *** for *p* < 0.001.

**Table 3 brainsci-12-00882-t003:** The comparison of left and right diaphragm mobility, thickness, and thickening fraction of healthy control, left hemiplegic, and right hemiplegic patients.

	Left Side	Right Side	*t*/*Z*	*p*
Healthy control (*n* = 20)				
Mobility (quiet breath)	1.48 ± 0.43	1.61 ± 0.45	1.476	0.156
Mobility (deep breath)	4.63 ± 1.14	5.26 ± 1.34	−3.179 ^c^	0.001 ***
Thickness (end-inspiratory)	0.32 ± 0.09	0.33 ± 0.09	0.604	0.553
Thickness (end-expiratory)	0.20 ± 0.05	0.21 ± 0.05	1.211	0.241
TF (%)	60.52 ± 31.83	58.05 ± 33.21	−0.16 ^b^	0.872
Left hemiplegia patients (*n* = 27)				
Mobility (quiet breath)	1.27 ± 0.50	1.41 ± 0.49	−1.33	0.195
Mobility (deep breath)	3.01 ± 1.26	4.41 ± 1.30	−4.397 ^b^	0.000 ***
Thickness (end-inspiratory)	0.32 ± 0.12	0.34 ± 0.10	−1.268	0.216
Thickness (end-expiratory)	0.23 ± 0.07	0.23 ± 0.06	−0.27	0.789
TF (%)	39.26 ± 44.22	48.59 ± 29.74	−2.114 ^b^	0.034 *
Right hemiplegia patients (*n* = 18)				
Mobility (quiet breath)	1.48 ± 0.53	1.39 ± 0.60	−0.598	0.558
Mobility (deep breath)	3.44 ± 1.55	3.40 ± 1.85	−0.134	0.895
Thickness (end-inspiratory)	0.32 ± 0.10	0.30 ± 0.08	−0.879 ^b^	0.379
Thickness (end-expiratory)	0.24 ± 0.07	0.24 ± 0.06	−0.871 ^c^	0.384
TF (%)	35.68 ± 22.68	23.48 ± 19.11	−2.243 ^b^	0.025 *

Data expressed as mean ± SD. TF (%), thickening fraction, Thickness and TF (%) were measured under deep breathing. *t* represents the result of *t*-test, *Z* represents the result of Wilcoxon signed ranks test. b, based on negative rank; c, based on positive rank. The value of *p* < 0.05 was considered significant, with symbols presenting as * for *p* < 0.05 and *** for *p* < 0.001.

**Table 4 brainsci-12-00882-t004:** The hemiplegic and non-hemiplegic diaphragm mobility and thickening fraction data of left and right hemiplegic patients.

	Left Hemiplegia Patients (*n* = 27)	Right Hemiplegia Patients (*n* = 18)	*t*/*Z*	*p*
Hemiplegic side				
Mobility (deep breath)	3.01 ± 1.26	3.40 ± 1.85	−0.869 ^a^	0.385
TF (%)	39.26 ± 44.22	23.48 ± 19.11	−1.274 ^a^	0.203
Non-hemiplegic side				
Mobility (deep breath)	4.41 ± 1.30	3.44 ± 1.55	2.268	0.028 *
TF (%)	48.59 ± 29.74	35.68 ± 22.68	1.561	0.126

Data expressed as mean ± SD. TF (%), thickening fraction (deep breath). *t* represents the result of *t*-test; *Z* represents the result of Wilcoxon signed ranks test. a, Wilcoxon signed ranks test. The value of *p* < 0.05 was considered significant, with symbol presenting as * for *p* < 0.05.

**Table 5 brainsci-12-00882-t005:** The result of multiple linear regression analysis coefficients ^a b c d^.

		Unstandardized Coefficients		Standardized Coefficients	*t*	*p*	*R*	Collinearity Statistics	
Model	Predictors	*b*	*SE*	*β*				*Tolerance*	*VIF*
1 ^a^	constant	0.273	0.334		0.818	0.418	0.544		
	Age	0.011	0.005	0.293	2.116	0.041		0.996	1.036
	FMA	0.009	0.003	0.407	2.938	0.006		0.996	1.036
2 ^b^	constant	2.444	0.389		6.290	0.000	0.331		
	Berg	0.024	0.011	0.331	2.221	0.032		1.000	1.000
3 ^c^	constant	0.233	0.032		7.169	0.000	0.390		
	FMA	0.002	0.001	0.390	2.648	0.012		1.000	1.000
4 ^d^	constant	0.185	0.019		9.741	0.000	0.452		
	FMA	0.001	0.000	0.452	3.164	0.003		1.000	1.000

^a^ Dependent variable, mobility under quiet breath; ^b^ Dependent variable, mobility under deep breath; ^c^ Dependent variable, thickness at end-inspiratory; ^d^ Dependent variable, thickness at end-expiratory. *B*, unstandardized coefficient; *SE*, standard error; *β*, standardized coefficient; *R*, correlation coefficient; *VIF*, variance inflation factor. FMA, Fugl–Meyer Motor Function Assessment score; Berg, Berg Balance Scale score.

## Data Availability

The data presented in this study are available upon reasonable request from the corresponding author.

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
