# Peer review of "Assessment of Diaphragm in Hemiplegic Patients after Stroke with Ultrasound and Its Correlation of Extremity Motor and Balance Function"

_brainsci, 2022, doi:10.3390/brainsci12070882_

Round 1
Reviewer 1 Report
The specific analysis carried out in this study regarding the motor behaviour of the diaphragm in this type of patient is interesting and provides new ideas for the future to complement Physiotherapy interventions more effectively. However, the functional implication of this diaphragmatic motor alteration has been lacking, and it would be interesting to incorporate it into future studies, especially focused on the activity and participation of the individual in his or her environment.
My suggestions are as follows:
1. Respiratory function may be related to gait, level of physical activity, no. of steps to improve.
2. Why is the n of the experimental group different from the control group? You should explain this in the limitations of the study.
3. Does relationship imply causality? If yes or no, this should be explained in the discussion.
4. In the inclusion criteria, the age range of the patients is very wide, could physiological ageing have something to do with the motor behaviour of the diaphragm in the older patients? You should reason this out and include it as a limitation in the discussion.
5. Wouldn't it be more interesting to perform the motor assessment of the diaphragm while the subjects perform a still postural activity such as sitting or standing? Could it be a line to investigate in the future that this type of assessment be done in functional activities?
Author Response
Response to Reviewer 1 Comments
The specific analysis carried out in this study regarding the motor behaviour of the diaphragm in this type of patient is interesting and provides new ideas for the future to complement Physiotherapy interventions more effectively. However, the functional implication of this diaphragmatic motor alteration has been lacking, and it would be interesting to incorporate it into future studies, especially focused on the activity and participation of the individual in his or her environment.
My suggestions are as follows:
Point 1: Respiratory function may be related to gait, level of physical activity, no. of steps to improve.
Response 1: There were several studies found the relationship between respiratory function and gait[1], level of physical activity[2]. We also found a positive correlation between diaphragm function and activities of daily living. And future studies can analyze the relationship between diaphragm function and those relevant factors mentioned above .
[1]Lee DK, Jeong HJ, Lee JS. Effect of respiratory exercise on pulmonary function, balance, and gait in patients with chronic stroke. J Phys Ther Sci. 2018 Aug;30(8):984-987.
[2]Kaneko H. Association of respiratory function with physical performance, physical activity, and sedentary behavior in older adults. J Phys Ther Sci. 2020 Feb;32(2):92-97.
Point 2: Why is the n of the experimental group different from the control group? You should explain this in the limitations of the study.
Response 2: We selected 20 age-matched healthy participants with normal lung function at the ratio of 2:1. The ratio between hemiplegic patients and healthy subjects was calculated based on the previously reported data and the final target sample size was a ratio of 1:1~3:1. And we have explained in the section of participant selection.(updated)
Point 3: Does relationship imply causality? If yes or no, this should be explained in the discussion.
Response 3: We described the correlation between diaphragm function and the extremity motor function, balance function of the hemiplegic side in the discussion. The relationship did not imply causality. And we have explained in the section of discussion. (updated)
Point 4: In the inclusion criteria, the age range of the patients is very wide, could physiological ageing have something to do with the motor behaviour of the diaphragm in the older patients? You should reason this out and include it as a limitation in the discussion.
Response 4: The function of the diaphragm and respiratory muscles decline to a certain extent with age [3,4]. It is a limitation that the inclusion criteria of the subjects was a wide age range. Future studies can expand the sample size and conduct a stratified study of diaphragm function in stroke patients of different ages. And we have explained in the section of discussion.(updated)
[3]Greising SM, Ermilov LG, Sieck GC, Mantilla CB. Ageing and neurotrophic signalling effects on diaphragm neuromuscular function. J Physiol. 2015 Jan 15;593(2):431-40.
[4]Bordoni B, Morabito B, Simonelli M. Ageing of the Diaphragm Muscle. Cureus. 2020 Jan 13;12(1):e6645.
Point 5: Wouldn't it be more interesting to perform the motor assessment of the diaphragm while the subjects perform a still postural activity such as sitting or standing? Could it be a line to investigate in the future that this type of assessment be done in functional activities?
Response 5: The supine position is preferred, because there is less overall variability, less side-to-side variability, and greater reproducibility[5,6]. Some researches showed that diaphragm excursion is known to be greater in the supine position for the same volume inspired than in the sitting or standing positions[7]. Therefore, in this study all participants used the supine position. In the future, after standardization of measures of diaphragm function in different positions, these different types of assessment may be attempted to be applied to functional activities.
[5]Gerscovich EO, Cronan M, McGahan JP, Jain K, Jones CD, McDonald C. Ultrasonographic evaluation of diaphragmatic motion. J Ultrasound Med. 2001 Jun; 20(6):597–604.
[6]Gierada DS, Curtin JJ, Erickson SJ, Prost RW, Strandt JA, Goodman LR. Diaphragmatic motion: Fast gradient-recalled-echo MR imaging in healthy subjects. Radiology. 1995 Mar; 194(3):879–884.
[7]Sarwal A, Walker FO, Cartwright MS. Neuromuscular ultrasound for evaluation of the diaphragm. Muscle Nerve 2013;47(3):319-29.

Reviewer 2 Report
Review of the manuscript -Manuscript ID: brainsci-1762965
The paper presents interesting studies important in the treatment of patients after stroke.
The title encourages to read the content of the article.
The summary is complete and meets the requirements.
The purpose of the publication was clearly defined.
The Method section presents some shortcomings.
In the Material and Method section, it would be nice to show the flow of participants in a diagram.
The study inclusion criteria already indicate a large discrepancy in the patients qualified for the examination (radiological and clinical diagnosis of ischemic or hemorrhagic stroke within 1-6 months after the onset of the disease, 2) between 30 and 80 years of age). Apart from that, the inclusion criterion No. 5) of voluntary participation in the study and signing the informed consent does not characterize, in terms of qualification for the study, the selected patients.
The authors did not provide inclusion and exclusion criteria for the control group. Please supplement.
Indeed, the limitation of the study is the number of people examined. Moreover, the authors did not take care of matching the studied groups, such as the ratio of women to men, and the average age of the respondents.
The average time from the onset of a stroke is approximately 3 months, but the standard deviation over a month indicates considerable discrepancies in this not-too-large group of respondents. It is well known that the duration and/or month of treating stroke patients change a lot in their functional evaluation. It would be helpful to get a closer look at the location of the stroke.
It would be interesting to clarify whether a respiratory infection could affect the side of the diaphragm (indirectly affected side 17.78% and infection 15.56%).
In the Discussion section, I would highlight the value of research. Moreover, after analyzing the above comments, please consider supplementing the study limitation.
The results of the study confirm the assumptions of the work.
Please analyze the proofreading of the language.
This study is important for the functional evaluation of stroke patients, especially since, as the authors themselves write, the study shows an association with other dysfunctions in stroke patients. The study may confirm the belief in the implementation of respiratory therapy in the described group of patients.
The work requires corrections before it is allowed to be published.
Author Response
Response to Reviewer 2 Comments
The paper presents interesting studies important in the treatment of patients after stroke.
The title encourages to read the content of the article.
The summary is complete and meets the requirements.
The purpose of the publication was clearly defined.
The Method section presents some shortcomings.
Point 1: In the Material and Method section, it would be nice to show the flow of participants in a diagram.
Response 1: We have added a Study protocol instead of a diagram in the Materials and Methods section: The participants were evaluated with the following assessments: diaphragm mobility and thickness with ultrasound under quiet and deep breathing, the general characteristics, and the assessments of extremity function. All the assessments were completed during a single visit. And we have added in the section of participant selection.(updated)
Point 2: The study inclusion criteria already indicate a large discrepancy in the patients qualified for the examination (radiological and clinical diagnosis of ischemic or hemorrhagic stroke within 1-6 months after the onset of the disease, 2) between 30 and 80 years of age). Apart from that, the inclusion criterion No. 5) of voluntary participation in the study and signing the informed consent does not characterize, in terms of qualification for the study, the selected patients.
Response 2: It was a limitation that the average duration of stroke in this study was 3 months that mainly in the convalescent period within 1-6 months from the onset. And the inclusion criteria of the subjects was a wide age range. We have explained in the discussion. And we removed inclusion criterion 5). (updated)
Point 3: The authors did not provide inclusion and exclusion criteria for the control group. Please supplement.
Response 3: Healthy subjects were screened and recruited as follows. We selected the age-matched healthy participants with normal lung function (FEV1 > 80% pred and FVC > 80% pred) according to the data of participants with stroke at the ratio of 2:1. And the exclusion criteria of the control group was same to the study group. And we have added in the section of participant selection.(updated)
Point 4: Indeed, the limitation of the study is the number of people examined. Moreover, the authors did not take care of matching the studied groups, such as the ratio of women to men, and the average age of the respondents.
Response 4: In the chapter of Sample Size Calculation, we described the sample size of the study group was calculated using data from our previous study, which showed the difference between the average mobility of the two groups was 0.5. We calculated that at least 34 patients were needed to provide the study with a sufficient statistical power of 0.9 and an alpha of 0.05. We selected 20 age-matched healthy participants with normal lung function (FEV1 > 80% pred and FVC > 80% pred) according to the age of stroke participants at the ratio of 2:1. The ratio between hemiplegic patients and healthy subjects was calculated based on the previously reported data and the final target sample size was a ratio of 1:1~3:1. And we have added the explanation in the section of participant selection. (updated)
There was a mismatch in control for the ratio of males to females at enrollment. And the inclusion criteria of the subjects was a wide age range. We have added the explanations in the discussion. (updated)
Point 5: The average time from the onset of a stroke is approximately 3 months, but the standard deviation over a month indicates considerable discrepancies in this not-too-large group of respondents. It is well known that the duration and/or month of treating stroke patients change a lot in their functional evaluation. It would be helpful to get a closer look at the location of the stroke.
Response 5: It was a limitation that the average duration of stroke in this study was 3 months that mainly in the convalescent period within 1-6 months from the onset. We have explained in the discussion. (updated)
The location of the stroke greatly affects both the dysfunction and recovery. In the study, the locations of the hemiplegic patients after stroke were as follows: Cortical 17 (37.78%), Subcortical 16 (35.56%), Cortical with Subcortical 11 (22.22%), brain stem 1 (2.22%). We compared the diaphragm function of the stroke at different location and found no significant difference. But due to the small sample size, the results need further verification.
Point 6: It would be interesting to clarify whether a respiratory infection could affect the side of the diaphragm (indirectly affected side 17.78% and infection 15.56%).
Response 6: In this study, 7 hemiplegia patients had pulmonary infection (15.56%). Among them, 3 patients with right hemiplegia had pulmonary infection (2 cases in the right pulmonary and 1 case in both), Among them, 0 cases of diaphragmatic dysfunction on the indirectly affected side. And 4 cases with left hemiplegia had pulmonary infection (1 case in the left pulmonary and 3 cases in both) ), Among them, 1 case of diaphragmatic dysfunction on the indirectly affected side. Pulmonary infection might be positively correlated with hemiplegic side of the stroke patients. It might be roughly judged that there was no correlation between pulmonary infection and indirectly affected side, but due to the small sample size, the results need further verification.
Point 7: In the Discussion section, I would highlight the value of research. Moreover, after analyzing the above comments, please consider supplementing the study limitation.
Response 7: We have explained the limitations above mentioned in the discussion. (updated)
The results of the study confirm the assumptions of the work.
Point 8: Please analyze the proofreading of the language.
Response 8: We have made a proofreading of the language.(Please see the attachment)
This study is important for the functional evaluation of stroke patients, especially since, as the authors themselves write, the study shows an association with other dysfunctions in stroke patients. The study may confirm the belief in the implementation of respiratory therapy in the described group of patients.
The work requires corrections before it is allowed to be published.

Reviewer 3 Report
The study tackles an interesting topic using low-cost and easy-to-use equipment. However, I believe it is appropriate to point out some problems that I encountered:
- English needs a massive revision. Some passages are really difficult to understand.
I believe that the major problems in terms of understanding the text are found in the chapter on materials and methods and in that of the discussion.
The type of study presented is a cross-sectional observational study, this should already be specified through the abstract. I believe it appropriate for the authors to refer to the STROBE guidelines for cross-sectional observational studies.
I did not find the potential confounders in the materials and methods.
Integrate possible study-related biases into materials and methods.
Explain how missing data were addressed in the statistical analysis.
I think the part of the discussion should be organized more clearly, some passages are redundant. I suggest focusing attention on the generalizability of the results and above all on the clinical impact that the results could have in the approach to this type of patient.
From a clinical point of view I would like to ask if it is possible to express the type of rehabilitation and how often it was carried out by the patients. In fact, this parameter may or may not influence the desired outcomes and above all also the parameters relating to the diaphragm.
Best regards
Author Response
Response to Reviewer 3 Comments
The study tackles an interesting topic using low-cost and easy-to-use equipment. However, I believe it is appropriate to point out some problems that I encountered:
Point 1: English needs a massive revision. Some passages are really difficult to understand.
Response 1: We have made a massive revision to the English.(Please see the attachment)
Point 2: I believe that the major problems in terms of understanding the text are found in the chapter on materials and methods and in that of the discussion.
Response 2: We have made a massive revision in the chapter of the materials and methods and the discussion. (Please see the attachment)
Point 3: The type of study presented is a cross-sectional observational study, this should already be specified through the abstract. I believe it appropriate for the authors to refer to the STROBE guidelines for cross-sectional observational studies.
Response 3: We have Indicated the study’s design with a commonly used term in the abstract. The type of this study was a cross-sectional observational study.(updated)
We have supplied The STROBE Statement—Checklist at the end of the manuscript.(Please see the attachment)
Point 4: I did not find the potential confounders in the materials and methods.
Response 4: We have matched the age of the included participants. Other potential confounders, including gender, duration of stroke were the limitations of this study. We have explained in the discussion. Multivariate analysis can analyze the effect of multiple factors on the outcome at the same time. Statistical analysis could partially control for the potential confounders.
Point 5: Integrate possible study-related biases into materials and methods.
Response 5: We have formulated the inclusion and exclusion criteria before the study. Among them, the age range was widely, and considering the characteristics of the long recovery period of stroke patients, and the duration of the disease was in the recovery period. These have been added in the discussion as limitations of the study.
In chapter of materials and methods, we standardize ultrasound machine, study conditions and methods. And we used the same observer for three different cycles of measurement, and a unified standard for measurement software, thus eliminating within-observer variability. All patients received assessments by an experienced physician from the rehabilitation medicine department. The above methods could partially control study-related biases, but there were still some limitations in this study.
Point 6: Explain how missing data were addressed in the statistical analysis.
Response 6: We excluded the patients who had partial data missing. We have added the explanation in the section of Results. (updated)
Point 7: I think the part of the discussion should be organized more clearly, some passages are redundant. I suggest focusing attention on the generalizability of the results and above all on the clinical impact that the results could have in the approach to this type of patient.
Response 7: We have reorganized the discussion. Some redundant passages were removed and we mainly focused on clinically relevant results. (updated) (Please see the attachment)
Point 8: From a clinical point of view I would like to ask if it is possible to express the type of rehabilitation and how often it was carried out by the patients. In fact, this parameter may or may not influence the desired outcomes and above all also the parameters relating to the diaphragm.
Response 8: There were many types of rehabilitation for stroke patients showed in previous research, including inspiratory muscle training[1], abdominal drawing-in maneuver[2], home-based respiratory muscle[3], game-based breathing exercise[4], et al. Through the analysis of literature, we found that the frequency of inspiratory muscle training was two daily sessions (consisting of two times 15 inspirations at normal breathing rhythm (5–10 min)), 7 days a week for 3 weeks training. The abdominal drawing-in maneuver was for 1 hour/day, five times a week for 6 weeks. The frequency of home-based respiratory muscle training was a 40-min of respiratory muscle training program, seven days/week, for eight weeks in their homes. Rehabilitation could increase strength, as well as improve respiratory muscles, pulmonary function and activity of stroke patients.
Our research group is also exploring the improvement of diaphragm function, respiratory pressure, extremity motor, balance and cardiorespiratory fitness(CRF) before and after the intervention of respiratory feedback training for stroke patients with diaphragmatic dysfunction.
[1] Sørensen SL, Kjeldsen SS, Mortensen SS, Hansen UT, Hansen D, Pedersen AR, et al. “More air—better performance—faster recovery”: Study protocol for randomised controlled trial of the effect of post-stroke inspiratory muscle training for adults. Trials 2021;22(1).
[2]Kim C, Lee J, Kim H, Kim I. Effects of the combination of respiratory muscle training and abdominal drawing-in maneuver on respiratory muscle activity in patients with post-stroke hemiplegia: A pilot randomized controlled trial. Top Stroke Rehabil 2015;22(4):262-70.
[3]Menezes KKPD, Nascimento LR, Polese JC, Ada L, Teixeira-Salmela LF. Effect of high-intensity home-based respiratory muscle training on strength of respiratory muscles following a stroke: A protocol for a randomized controlled trial. Braz J Phys Ther 2017;21(5):372-7.
[4]Song C. The effects of Game-Based breathing exercise on pulmonary function in stroke patients: A preliminary study. Med Sci Monitor 2015;21:1806-11.

Round 2
Reviewer 3 Report
Thank you for making the required changes. I am in favor of publishing the article.
Best Regards